# SSNdesign—An R package for pseudo-Bayesian optimal and adaptive sampling designs on stream networks

Alan R. Pearse[1,2]*, James M. McGree[2,3], Nicholas A. Som[4,5], Catherine Leigh[1,2,3¤], Paul Maxwell[6], Jay M. Ver Hoef[7], Erin E. Peterson[1,2,3]

**1** Institute for Future Environments, Queensland University of Technology, Brisbane, QLD, Australia, **2** Australian Research Council Centre of Excellence for Mathematical and Statistical Frontiers, Queensland University of Technology, Brisbane, QLD, Australia, **3** School of Mathematical Sciences, Queensland University of Technology, Brisbane, Australia, **4** US Fish and Wildlife Service, Arcata, CA, United States of America, **5** Humboldt State University, Arcata, CA, United States of America, **6** Healthy Land and Water, Brisbane, QLD, Australia, **7** Alaska Fisheries Science Center, NOAA Fisheries, Seattle, WA, Australia

¤ Current address: Biosciences and Food Technology Discipline, School of Science, RMIT University, Bundoora, Victoria, Australia
* arp320@uowmail.edu.au

**Data Availability Statement:** The data referred to in the text are available with the SSNdesign package on GitHub at https://github.com/apear9/SSNdesign.

## Abstract

Streams and rivers are biodiverse and provide valuable ecosystem services. Maintaining these ecosystems is an important task, so organisations often monitor the status and trends in stream condition and biodiversity using field sampling and, more recently, autonomous *in-situ* sensors. However, data collection is often costly, so effective and efficient survey designs are crucial to maximise information while minimising costs. Geostatistics and optimal and adaptive design theory can be used to optimise the placement of sampling sites in freshwater studies and aquatic monitoring programs. Geostatistical modelling and experimental design on stream networks pose statistical challenges due to the branching structure of the network, flow connectivity and directionality, and differences in flow volume. Geostatistical models for stream network data and their unique features already exist. Some basic theory for experimental design in stream environments has also previously been described. However, open source software that makes these design methods available for aquatic scientists does not yet exist. To address this need, we present `SSNdesign`, an R package for solving optimal and adaptive design problems on stream networks that integrates with existing open-source software. We demonstrate the mathematical foundations of our approach, and illustrate the functionality of `SSNdesign` using two case studies involving real data from Queensland, Australia. In both case studies we demonstrate that the optimal or adaptive designs outperform random and spatially balanced survey designs implemented in existing open-source software packages. The `SSNdesign` package has the potential to boost the efficiency of freshwater monitoring efforts and provide much-needed information for freshwater conservation and management.

**Funding:** This study received funding and support from Healthy Land and Water (https://hlw.org.au/) and was motivated by their monitoring needs and desire to transition their freshwater monitoring program to the use of in-situ sensors at broad spatial scales. E.E.P and J.M.M received the award from Healthy Land and Water. P.M. works for Healthy Land and Water and P.M. assisted in identifying the two motivating examples / case studies for the study and provided the data used in the second case study. P.M. also contributed to the preparation of the manuscript. In addition, JMM was supported by an Australian Research Council Discovery Project (DP200101263).

**Competing interests:** The authors have declared that no competing interests exist.

# Introduction

Streams and rivers are highly biodiverse ecosystems supporting both aquatic and terrestrial species [1, 2] and provide important ecosystem services including clean water, food, and energy [3]. The ecological and economic importance of waterways has driven government and non-government organisations worldwide to invest large amounts of time and money into their monitoring, assessment and rehabilitation [4]. However, monitoring data remain relatively sparse [5] because the cost of sampling makes it impossible to gather data everywhere, on every stream, at all times. Thus, it is crucial to select sampling locations that yield as much information as possible about water quality and aquatic ecosystem health, especially when the stream system is large and resources for sampling are limited.

Geostatistical models are commonly used to analyse environmental data collected at different locations and to make predictions, with estimates of uncertainty, at unobserved (i.e. unsampled) sites [6]. These models are a generalisation of the classic linear regression model, which contains a deterministic mean describing the relationship between the response (i.e. dependent variable) and the covariates (i.e. independent variables). In a geostatistical model, the assumption of independence of the errors is relaxed to allow spatial autocorrelation, which is modelled as a function of the distance separating any two locations [7]. This provides a way to extract additional information from the data by modelling local deviations from the mean using the spatial autocorrelation, or covariance, between sites. However, spatial autocorrelation may exist in streams data that is not well described using Euclidean distance, given the branching network structure, stream flow connectivity, direction and volume [8]. In addition, many traditional covariance functions are invalid if an in-stream (i.e. hydrologic) distance measure is substituted for Euclidean distance [4, 9]. The use of covariance functions based on Euclidean distance may produce physically implausible results; for example, implying that two adjacent streams that do not flow into each other and that have separate watersheds are strongly related. This led to the development of covariance functions that are specifically designed to describe the unique spatial relationships found in streams data [4, 10]. Geostatistical models fit to streams data describe a number of in-stream relationships in a way that is scientifically consistent with the hydrological features of natural streams and, as such, are increasingly being used for broad-scale monitoring and modelling of stream networks; see, for example, Isaak et al. [11] and Marsha et al. [12], both model temperature in streams, with Marsha et al. [12] further considering questions of site placement and sample size based on their data.

The theoretical properties of geostatistical models can also be exploited in optimal and adaptive experimental designs [13–16], which are used to select sampling locations that maximize information gain and minimize costs. However, the exact locations included in an optimal design will depend on the objectives of the monitoring program. Common objectives include estimating the parameters of the underlying geostatistical model (e.g. fixed effects estimates and/or covariance parameters), making accurate predictions at unsampled locations, or both. Utility functions are mathematical representations of the objectives used to measure the suitability of a design for a specific purpose. Depending on the objective of the sampling, the best design might be one that includes spatially balanced sites distributed across the study area or it could be a design that includes clusters of sites in close proximity to one another [17]. A variety of utility functions are available [15, 16] and are described more specifically in Section 2.4. An adaptive design (i.e. sequential design) is constructed by making a series of optimal decisions about where to put sampling sites as new information becomes available over time [14]. For example, the spatial location of monitoring sites may change through time, with some sites removed due to changes in access, or additional sites added as new funding

becomes available. In these situations, the information gained from the data collected up to that point can be used to inform where the optimal sampling locations will be at the next time step. Hence, adaptive designs may provide additional benefits for long-term environmental monitoring programs because one-off optimal designs ignore the evolving nature of environmental processes and do not allow for adjustments as monitoring needs change [17].

Bayesian and pseudo-Bayesian methods can enhance optimal and adaptive designs. Utility functions often depend on the parameters of the geostatistical model that one intends to fit over the design; however, the utility function can only be evaluated when these parameter values are fixed [13]. If the values change, for example, through random variations in field conditions, then the design may no longer be optimal. Bayesian and pseudo-Bayesian optimal design addresses this issue by using simulation to construct more robust designs and to incorporate prior information about the distribution of the model parameters when constructing the design [18]. A drawback is that these methods are computationally intensive [18, 19]. In SSNdesign, we use the pseudo-Bayesian approach. This is different than a fully Bayesian approach because we are committed to performing frequentist inference on the data we collect from an experiment. The pseudo-Bayesian approach also does not take a Bayesian view of uncertainty, particularly with respect to model uncertainty, and as such we do not always have access to Bayesian utilities. Nevertheless, the pseudo-Bayesian approach allows us to incorporate prior information in the design process, which is not possible for purely frequentist designs, and can be more computationally efficient than the fully Bayesian approach.

Although numerous software packages have been developed to implement geostatistical models on streams and to solve experimental design problems, none have done both. The SSN package [20] for R statistical Software [21] is currently the only software available for implementing geostatistical models on stream networks [10]. However, various software packages exist to solve experimental design problems. For example, acebayes provides an implementation of the approximate coordinate exchange algorithm for finding optimal Bayesian designs given a user-specified utility function [22]. For spatial design problems, spsurvey [23] implements a variety of sampling designs including the Generalised Random Tessellation Sampling (GRTS) design for spatially balanced samples [24]. The package geospt [25] focuses on drawing optimal and adaptive spatial samples in the conventional 2-D geostatistical domain, with Euclidean distance used to describe spatial relationships between locations. However, it does not allow for stream-specific distance measures and covariance functions or it does not calculate design criteria consistent with a Bayesian or pseudo-Bayesian approach. This is important for constructing designs that are robust to changes in parameter values that the utility function depends on. Som et al. [15] and Falk et al. [16] made use of the geostatistical modelling functions in the SSN package for solving design problems on stream networks, but neither addressed adaptive design problems and both used customised code that was not made publicly available.

To our knowledge, SSNdesign is the first software package that allows users to optimise experimental designs on stream networks using geostatistical models and covariance functions that account for unique stream characteristics within a robust design framework. It combines the statistical modelling functionality found in the SSNpackage [20] with principles of pseudo-Bayesian design [13] into a generalised toolkit for solving optimal and adaptive design problems on stream networks. In Section 2, we discuss the mathematical principles underpinning these tools and outline the structure of the core functions in SSNdesign package along with a summary of the package's speed and performance. This section is extended by S1 Appendix, which gives a deeper treatment of the required mathematics. In Section 3 we present two case studies using real data from Queensland, Australia. S2 Appendix is the package vignette, and provides the reader with detailed code required to reproduce the example. We conclude with a

brief discussion of the package and future developments in Section 4. A glossary of terms is also provided in S3 Appendix for those unfamiliar with the language of experimental design.

## The SSNdesign package

### Software and data availability

The SSNdesign package is publicly available at https://github.com/apear9/ SSNdesign, along with the data used in this paper. This R package requires R version 3.5.0 or later, and depends on the packages SSN [20], doParallel [26], doRNG [27], spsurvey [23], and shp2graph [28]. These are downloaded from CRAN during the installation of the package.

### Workflow

There are three different workflows in SSNdesign corresponding to different design types, as well as for importing and manipulating stream network data (Figs 1 and 2). The general process is to import a stream dataset (see Section 2.2) and if necessary, create potential sampling locations and simulate data at those locations. The next step is to use the main workhorse function optimiseSSNDesign to find optimal or adaptive designs, or use drawStreamNetworkSamples to find probability-based designs. In the case of adaptive design problems, there are always multiple 'timesteps'. The first timestep is important because adaptive designs cannot be constructed without a pre-existing design. Therefore, it is not possible to go straight from data processing to an adaptive design; as such, the first design decision must be based on either an optimal or probability-based design before implementing the adaptive design process in subsequent timesteps (Fig 2).

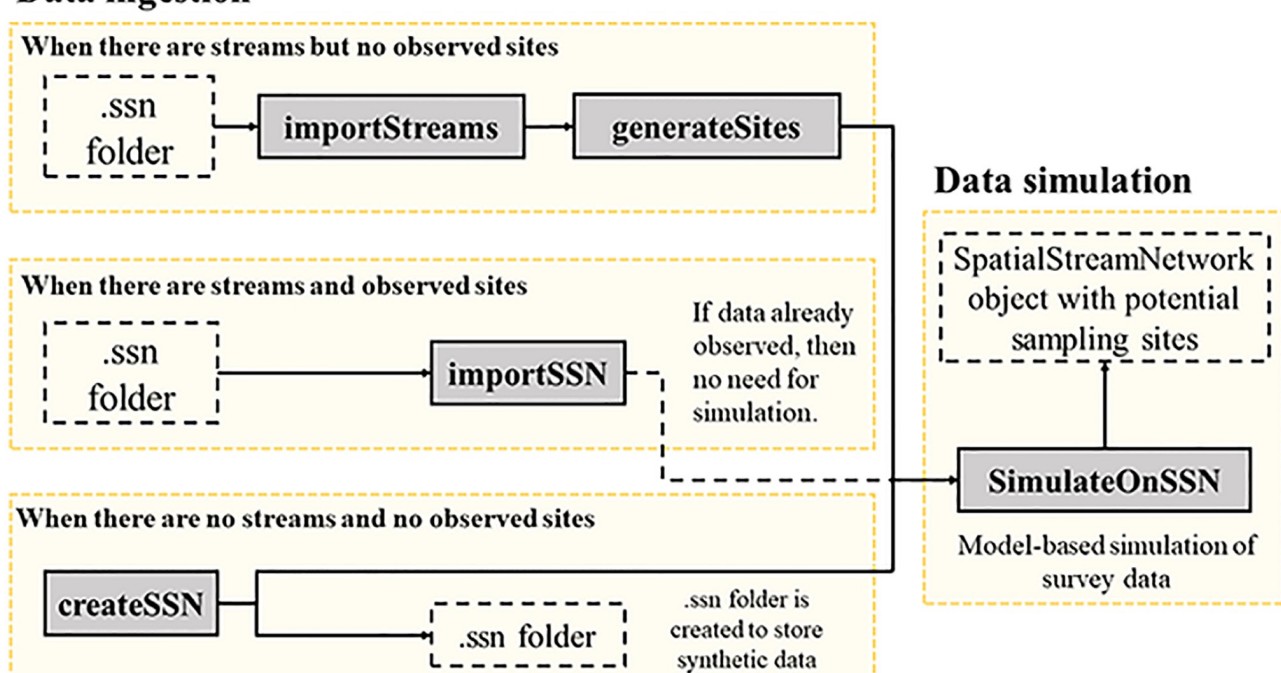

**Fig 1. A flow chart of the function calls used to import and prepare streams data for use in SSNdesign.** Grey boxes with solid outlines indicate a call to a function (N.B. importSSN and SimulateOnSSN belong to SSN, not SSNdesign). Clear boxes with dashed outlines indicate a file, folder or R object that is created as a result of a function call. The one dashed line represents an action that will not always be necessary.

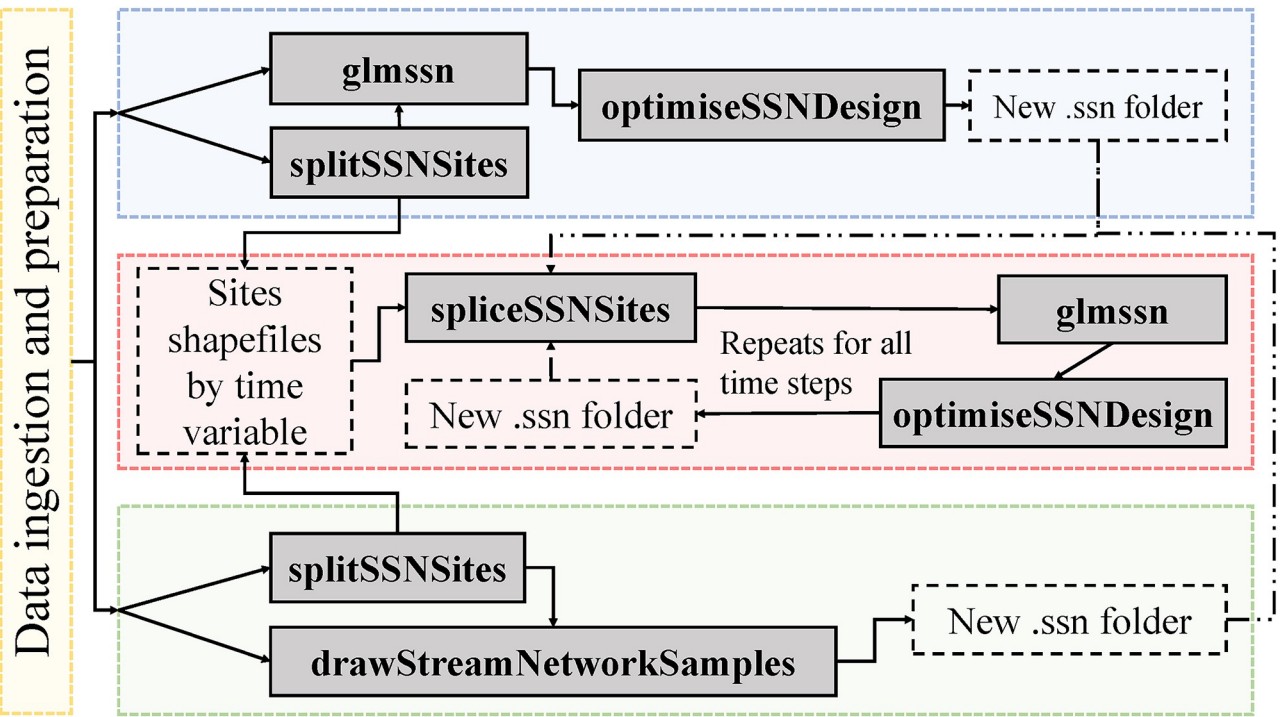

**Fig 2. A flow chart of the function calls used to construct optimal (blue), adaptive (red), and probability-based (green) designs for stream networks.** The yellow box is discussed in more detail in Fig 1. Grey boxes with solid outlines represent a call to a function (N.B. `glmssn` belongs to `SSN`, not `SSNdesign`). Clear boxes with dashed outlines indicate a file, folder or R object that is created as a result of a function call. The dash-dot-dot lines represent optional steps or connections between function calls that do not always occur.

## Data format, ingestion and manipulation

The `SSNdesign` package builds on the functionality in the package SSN [20], most notably the S4 `SpatialStreamNetwork` class for stream network datasets and the function and class `glmssn` which fits and stores fitted spatial stream network models. `SpatialStreamNetwork` objects are an extention of the `sp` class, but are unique because they contain a set of stream lines (i.e. edges) and a set of observed sites. Prediction sites can also be included, but are optional. These spatial data are imported from.ssn folders, which are created using the Spatial Tools for the Analysis of River Sytems (STARS) ArcGIS custom toolset [29]. The `importSSN` found in the SSN package is used to ingest data contained in the.ssn folders, but this function will not work if there are no observed sites in the.ssn folder. Therefore, the `SSNdesign` package provides the function `importStreams` for creating a `SpatialStreamNetwork` object with no observed or predicted sites. Additional functions such as `generateSites` are provided to add potential sampling sites to these empty networks. Four distinct workflows for importing and preprocessing stream data are shown in Fig 1, but the end result is a `SpatialStreamNetwork` object that contains streams and observed sites where data were collected or simulated that can be used for optimal or adaptive designs.

The data processing (Fig 1) and design (Fig 2) workflows produce objects of class `ssndesign`, which are lists containing 1) information about the way the design optimisation function (`optimiseSSNdesign`) was used, including the `SpatialStreamNetwork` objects before and after an optimal design is found, 2) data about the optimal or adaptive design, and 3) diagnostic information about the optimisation procedure. This class also has a plot method, `plot.ssndesign`, which plots the trace for the optimisation algorithm. The method

`plot.SpatialStreamNetwork` from the package `SSN` can be used to visualise the locations of the selected sites. Further details are provided in the package vignette (S2 Appendix). The vignette is intended as a practical guide that demonstrates the functionality and workflow of the package and as a practical reference for managers.

## Expected utility estimation and maximisation

There are many ways to configure a fixed number of monitoring sites. Each potential configuration represents a 'design' *d*, and the set of all possible configurations is denoted as *D*. The goal of optimal design theory as applied to stream networks is to find which configuration of sites is most suitable to achieve a purpose (e.g. precise parameter estimation in a statistical model). We refer to the optimal configuration of sites as $d^*$. Inside `SSNdesign`, the quality of a design and its suitability for a stated goal is measured using a function called the expected utility $U(d)$ [13]. Larger values of $U(d)$ indicate better designs and the calculation of $U(d)$ is linked to the utility function, $U(d, \theta, y)$. The utility function may depend on elements of the geostatistical model fitted over the design, including its parameters $\theta$ and either observed or predicted data *y* [13]. However, in many cases of pseudo-Bayesian utility functions, the utility function does not depend on *y* and can be written $U(d, \theta)$ [16]. This function mathematically encodes the criterion used to compare designs. Examples of utility functions can be found in Section 2.1.2. The utility function, however, cannot be used directly to assess the quality of designs. This is because $U(d, \theta, y)$ depends on specific values of $\theta$ and *y* and the relative rankings of designs may change, sometimes dramatically, if there are small variations in these two quantities. Therefore, the parameters $\theta$ and the data *y* must be integrated out such that the values used to rank designs depend only on the designs themselves. We achieve this using Monte-Carlo integration [13]. Further details are provided in S1 Appendix.

The set of possible designs *D* is usually large and, due to time and computational constraints, we cannot find $d^*$ by evaluating $U(d)$ for every $d \in D$. `SSNdesign` deals with this problem in two ways. Firstly, we do not treat the design problem as a continuous one. That is, we do not allow sites to shift to any place along the stream edges during the search for the best design. The user must first create a set of *N* candidate points, and a design containing *n* points is chosen from among them. This ensures that *D* has a finite size. Secondly, we reduce the computational load of finding $d^*$ [19] by applying a coordinate exchange algorithm called the Greedy Exchange Algorithm (S1 Appendix, Algorithm 1). This algorithm rapidly converges on highly efficient designs, although this efficient design may not be the best design [30]. Note that the Greedy Exchange Algorithm has previously been used for optimal designs on stream networks [16].

## Utility functions for optimal and adaptive experimental designs

The utility functions implemented in the `SSNdesign` package are suitable for solving either static optimal design problems or evolving, adaptive design problems. For optimal design, there are six utility functions for common monitoring objectives including parameter estimation and prediction (Table 1; S1 Appendix). For adaptive design, there are three utility functions for similar monitoring objectives that are appropriate for adaptive decision-making. These are intended to be used with the function `optimiseSSNDesign`. We also provide two utility functions for finding space-filling designs (Table 1), using the optimisation function `constructSpaceFillingDesigns`. These designs contain roughly equally spaced and unclustered sets of monitoring sites along the stream network [31, 32].

To solve adaptive design problems, we use a myopic design approach; that is, when making an adaptive design decision at a given point in time, we try to find the best decision for the

**Table 1. Utility functions implemented in `SSNdesign`. Empirical utility functions are utility functions where the covariance parameters are estimated from data simulated using the prior draws.** $\theta$ = a vector of covariance parameters from a geostatistical model; and $y$ = data that is either directly observed from a process or simulated from it. OP = optimal design; AD = adaptive design. n/a = no covariance parameters involved. $I(\theta)$ = the expected Fisher Information Matrix; $\hat{\beta}_{gls}$ = the estimates of the fixed effects; $Var(\hat{\beta}_{gls})$ = covariance matrix for the fixed effects. $s_z$ = a prediction site; $S$ = the set of all prediction sites. $\hat{y}(s_z)$ = the predicted value at a prediction site. $Var(\hat{y}(s_z))$ = the kriging variance. $O_t(\theta)$ = a summary statistic from the existing design. $D(x_i, x_j)$ = the distance between two points $x_i$ and $x_j$. The distance can be measured as Euclidean distance or hydrological distance along the stream network [10]. $D$ = a sorted vector of non-zero distances in a distance matrix; $J$ = the number of times each distance occurs in one triangle of the matrix. The subscript $w = 1, 2, \ldots, W$ counts the $W$ unique non-zero entries in the distance matrix. $p$ = a weighting power, with $p \geq 1$. In the Empirical column, $\times$ means No, $\checkmark$ means Yes.

| Name | Purpose | Application | Empirical | Definition of the expected utility | Reference |
|------|---------|-------------|-----------|------------------------------------|-----------|
| CP-optimality | Covariance parameters | OP | $\times$ | $\log \det[I(\theta)^{-1}]$ | [16] |
| D-optimality | Fixed effects parameters | OP | $\times$ | $\log \det [Var(\hat{\beta}_{gls})^{-1}]$ | [15, 16] |
| ED-optimality | Fixed effects parameters | OP | $\checkmark$ | $\log \det [\hat{Var}(\hat{\beta}_{gls})^{-1}]$ | [15, 16] |
| CPD-optimality | Fixed effects and covariance parameters, a mixture of CP- and D-optimality | OP | $\times$ | $\log \det [Var(\hat{\beta}_{gls})^{-1}] + \log \det [I(\theta)^{-1}]$ | |
| K-optimality | Predictions | OP, AD | $\times$ | $\left(\sum_{s_z \in S} Var(\hat{y}(s_z))\right)^{-1}$ | [15, 16] |
| EK-optimality | Predictions | OP, AD | $\checkmark$ | $\left(\sum_{s_z \in S} \hat{Var}(\hat{y}(s_z))\right)^{-1}$ | [15, 16] |
| Sequential CP-optimality | Covariance parameters | AD | $\times$ | $\log \det[(I(\theta) + O_t(\theta)^{-1}]$ | S1 Appendix |
| Sequential D-optimality | Fixed effects parameters | AD | $\times$ | $\log \det [(Var(\hat{\beta}_{gls}) + O_t(\theta))^{-1}]$ | S1 Appendix |
| Sequential ED-optimality | Fixed effects parameters | AD | $\checkmark$ | $\log \det [(\hat{Var}(\hat{\beta}_{gls}) + O_t(\theta))^{-1}]$ | S1 Appendix |
| Maximin | Space-filling, with an emphasis on increasing the minimum distance between pairs of points | OP | n/a | $\min_{i \neq j} D(x_i, x_j)$ | [32] |
| Morris-Mitchell | Space-filling, with an emphasis on increasing separations larger than the minimum | OP | n/a | $-\left(\sum_{w=1}^{W} (J_w D_w)^p\right)^{1/p}$ | [31] |

next time step only [33]. An alternative method is to use backward induction, which involves enumerating all possible future decisions for a set number of time steps and then choosing the best series of decisions from among them [14]. However, this approach is often computationally prohibitive. Here we use an algorithm that assumes an initial design $d_0$ that we seek to improve by adding sites or by rearranging the existing ones. We know *a priori* that, instead of solving this problem once, we will have to make a further $T$ decisions in the future about the arrangement of the design points (Algorithm 2, S1 Appendix). At each step $t = 1, 2, \ldots, T$ in this iterative process, we fit a model to the existing design at $d_{t-1}$ and summarise the fit of the model using the summary statistic $O_t(\theta)$. The summary statistic can be arbitrarily defined, though it should be low-dimensional and preferably fast to compute [14]. In evaluating the designs at each timestep $t$, we incorporate the summary statistic $O_{t-1}(\theta)$ in the calculation of the expected utility given the previous design decisions $U(d|d_{0:t-1}, y_{0:t-1})$. Data simulated from the likelihood at time $t$, $p(y|\theta^t, d)$, can involve real data collection. However, for most design studies, this step will simply involve generating predictions, with associated errors, from an assumed model. As with optimal design problems, the best design $d_t^*$ at step $t$ is the one which maximises $U(d|d_{0:t-1}, y_{0:t-1})$. We also update the priors on the parameters for each new design. The process is repeated until the best design has been found for each of $T$ time steps. For the objectives of covariance parameter estimation and fixed effects parameter estimation, we have defined utility functions specifically. Note that the main difference between the adaptive and static utility functions is the use of the summary statistic $O_t(\theta)$ from the model fitted to the existing sites (Table 1; Supplementary Information A). These utility functions can be used with `optimiseSSNDesign`.

Users may also define their own utility functions since the `optimiseSSNDesign` function has the flexibility to accept utility functions as an argument. The utility function must be defined in this format: `utility.function(ssn, glmssn, design, prior.parameters, n.draws, extra.arguments)`. The exact requirements in terms of input type and additional data accessible within the function `optimiseSSNDesign` are described in the function documentation. It is not necessary to use all the arguments inside the function. Ultimately, the only requirement for a working user-defined utility function is that the function returns a single number, representing the expected utility.

## Other standard designs

The `SSNdesign` package focuses on optimal and adaptive design problems, but we also include a number of standard designs such as simple random sampling and GRTS [24]. We have also included heuristic sampling schemes designed specifically for stream networks, such as designs with sites allocated to headwater streams (i.e. streams at the top of the network), to outlets (i.e. most downstream location on the network), or in clusters around stream confluences (i.e. junctions where stream segments converge; Som et al., [15]). These are all options for the function `drawStreamNetworkSamples` (Table 2).

## Computational performance

The optimisation of Bayesian and pseudo-Bayesian experimental designs via simulation is notoriously slow [19]. Other experimental design packages explicitly warn users to expect this (e.g. acebayes [22]). In our case, we have parallelised the functions (compatible with any OS) to increase computational efficiency for large problems. However, in many common situations users can expect run-times of hours, to days, and even weeks. The expected computation time in hours is given by $K \times L \times T \times n \times (N - n)/(3600 \times C)$, where $K$ is the number of random starts that are used to seed the algorithm, $L$ is the number of times the algorithm must iterate before converging, $T$ is the time in seconds that is required to calculate $U(d)$ for a single design, $n$ is the number of desired sampling locations, $N$ is the number of potential sampling locations, and $C$ is the number of CPUs allocated to the task. The parameter $K$ is specified by the user in `optimiseSSNDesign`, but $L$ is more difficult to constrain. The number of times the algorithm must iterate until convergence is stochastic but, in our experience, $L = 2$ is most common; though we have observed $L \in \{3, 4, 5\}$. Unsurprisingly, the number of potential sampling locations and the number of desired sampling locations

**Table 2. Standard designs from Som et al. [15].** Note that 'name in package' is the string argument that must be passed to the `drawStreamNetworkSamples` function to use the sampling scheme.

| ID | Name in package | Description |
|---|---|---|
| SRS | SRS | Simple random sampling. An unstratified random sample of sites. |
| G1 | GRTS | GRTS. Spatially balanced design [24]. |
| G2 | GRTSmouth | GRTS with one site always assigned to the stream outlet. |
| G4 | GRTSclus | GRTS with clusters of sites around confluences. |
| H1 | Headwater.Clusts.and. Singles | Headwater samples. One site allocated to the outlet and others preferentially allocated to tributaries. |
| C3 | Trib.Sets.Head.Singles. sample | Clustered triplets. As many points as possible are allocated to triplets clustered where each segment meets at a confluence. All remaining points are assigned to tributaries. |
| C4 | Trib.Sets.Head.Singles. Mouth.sample | C3 with a single point allocated to the outlet segment. |

strongly influence computing time, and these make the largest contribution when $n \approx N/2$. Large problems $N \geq 300$ generally take at least a day, if not more, to compute. Our second case study below with $N = 900$ required approximately 4 days to complete using 20-32 CPUs and $n \ll N$. Using complicated or intensive utility functions will also add significantly to computation time. The empirical prediction or estimation utilities are particularly prone to computational inefficiencies because they rely on iteratively fitting and predicting from geostatistical stream network models. This is compounded by the fact that larger $n$ is often associated with larger $T$ values, due to the increased size of the matrices (e.g. covariance matrix, design matrix) that must be created and/or inverted.

## Case studies

We present two case studies where we illustrate how SSNdesign can be used to address common monitoring challenges. Field data collection can be expensive, so existing monitoring programs often need to reduce sampling effort due to resource constraints. In the first case study, we demonstrate how to ensure that the remaining survey sites are optimally placed to minimize information loss when sampling effort is reduced. In the second case study, we show how adaptive sampling can be used to design a monitoring program in a previously unsampled area, by gradually adding additional survey sites year-by-year. The R code used to create these examples is provided in Supplementary Information B so that readers can re-create them and also apply these methods to their own data.

In a model-based design problem, a 'true model' must be specified. Here, a true model refers to the statistical model which most adequately characterises the underlying spatial process given what we know about the system. If a historical dataset from the study area exists, a standard model-selection process can be used to determine which model has the most support in the data using a range of approaches (e.g. information criteria or cross validation). If historical data are not available, simulated data can also be used to implement the model-based design. In this case, the general approach is to:

1. Specify the form of the statistical model;

2. Identify which potential covariates should be related to the response (e.g. temperature affects the solubility of dissolved oxygen in water) based on prior knowledge of the system, or similar systems; and

3. Set priors that specify the likely relationship between covariates and the response based on previously collected data, expert opinion and/or a literature review.

The same process must be undertaken to specify the spatial covariance structure and covariance parameters for the model. The SimulateOnSSN function from SSN [20] can then be used to simulate data that are subsequently used to implement the model-based design. Needless to say, the quality of the design will depend strongly on the quality of the prior information.

### Case study 1: Lake Eacham

Water temperature samples were collected at 88 sites along a stream network near Lake Eacham in north Queensland, Australia (Fig 3) [34]. The dataset includes a shapefile of streams, the 88 observed sites with rainfall data and GIS-derived urban and grazing land use percentages in the catchment, and 237 prediction sites with the same covariates. Most survey sites were clustered at stream confluences, with multiple sites located in the upstream and downstream segments. In a similar experiment to that of Falk et al. [16], we used optimal experimental design to reduce

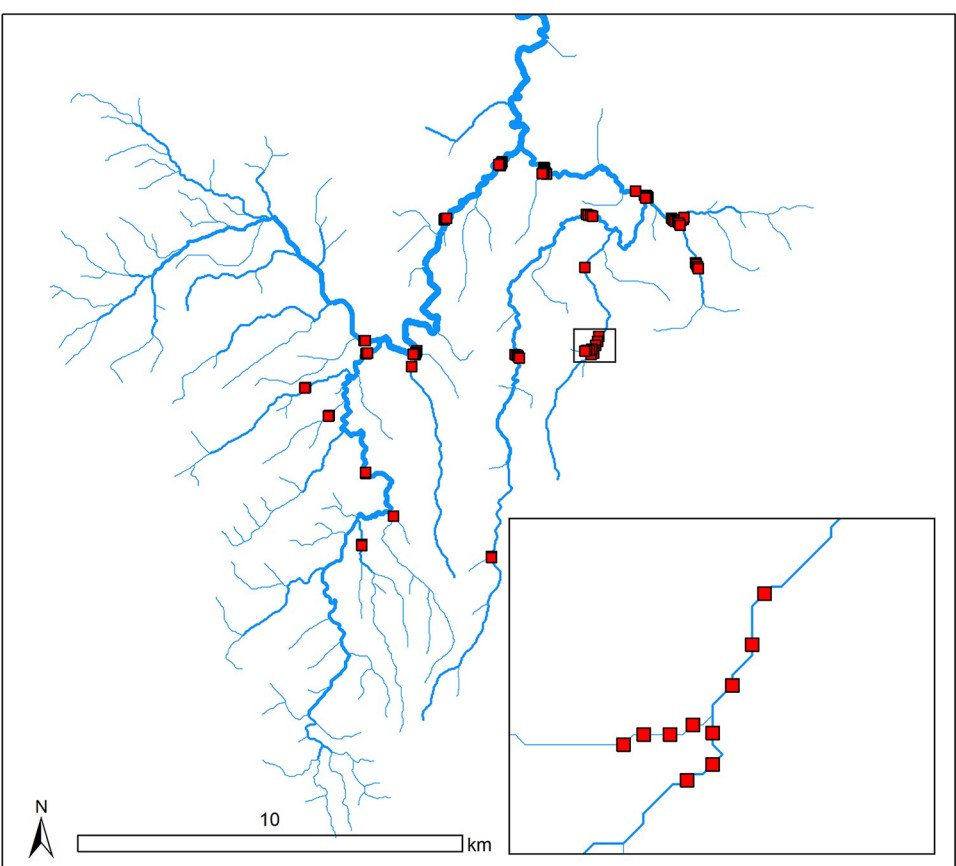

**Fig 3. The Lake Eacham stream network (blue lines) with all potential sampling sites (red squares).** The width of the stream lines is proportional to the catchment area for each stream segment. The inset shows the spatial distribution of sampling sites around a confluence in the stream network.

the number of survey sites by half, with the least amount of information loss possible. More specifically, we wanted to retain the ability to 1) estimate the effect that land use and rainfall are having on water temperature and 2) accurately predict temperature at unsampled locations. In the former case, we optimised the design using CPD-optimality, which is maximised when uncertainties in both the fixed-effects and covariance parameters are minimised (Table 1). In the latter case, we used K-optimality, which is maximised when the total prediction uncertainty across all sites is minimised. Note that this situation where two designs are built separately using different utility functions is not ideal. There is no way to reconcile the two resulting designs into a single one. Ideally, we would be able to use the EK-optimality function. This utility function aims to maximise prediction accuracy but also involves a parameter estimation step, and therefore serves as a dual purpose utility function that yields designs that are efficient for parameter estimation and prediction. However, it was not practical to use the EK-optimality function because it is extremely computationally expensive, even for this dataset with only 88 potential sites.

We fit a spatial stream-network model to the temperature data, using riparian land use (i.e. percent grazing and urban area) and the total rainfall recorded on the sampling date (mm) as covariates. The covariance structure contained exponential tail-up and tail-down components [10]. Log-normal priors were set on the covariance parameters using the natural logarithm of

the estimated covariance parameters as the means and the log-scale standard errors of {0.35, 0.56, 0.63, 0.69, 0.68}, which were also estimated from the existing data using restricted maximum likelihood (REML).

We set about finding CPD- and K-optimal designs with 44 sites by removing one site at a time from the original 88 sites. An alternative approach would be to optimise for a 44 site design with no intermediate steps. However, we chose to remove one site at a time because it helps reveal differences in the decision process between the CPD- and K-optimality utility functions that would otherwise be difficult to identify. Note that this is not an adaptive design because the model is not refit and the priors on the covariance parameters are not updated at each step. The expected CPD- and K-optimal utilities were calculated at each step using 500 Monte Carlo draws.

As expected, the results revealed differences between the intermediate and final designs discovered under CPD- and K-optimality. Both utility functions preserved clusters in groups of at least three sites around confluences (one on each of the upstream segments and one on the downstream segment). However, CPD-optimality appeared to remove single sites that were not part of clusters in preference to sites within clusters. By comparison, K-optimality appeared to reduce clusters around confluences down to three sites much more quickly, while preserving sites that were located away from confluences. This reflects the previously observed tendency of designs constructed for parameter estimation to favour spatial clusters and the tendency of designs that optimise prediction to contain sites spread out in space, with some clusters to characterise spatial dependence at short distances [15, 16]. These results suggest that clusters located around confluences provide valuable information about the covariance structure that is needed to generate precise parameter estimates and accurate predictions.

We tracked the information loss from the design process over time and compared the performance of our final 44-site optimal designs against 20 random and GRTS designs of the same size (Fig 4). We compared them to multiple GRTS and random designs because these designs have many potential configurations. Therefore, we needed to characterise the range of their performances under the chosen expected utilities. Information in this context is measured as relative design efficiency; a ratio of the expected utility of a given design and the expected utility of the 'best' design, which in this case contained all 88 sites. There was a linear reduction of information available for parameter estimation as sites were eliminated from the design (Fig 4a). While the optimal design containing 44 sites provided only 20% of the information gained from 88 sites, it provided significantly more information than the random and GRTS designs. These results suggest that reducing sampling effort by 50% would signficantly impact parameter estimation. However, the same was not true for prediction. We observed only minor reductions in the efficiency of the optimal 44-site design compared to the full 88-site design, while also demonstrating considerable gains in efficiency over random and GRTS designs (Fig 4b). As such, a 50% reduction in sampling effort would have little impact on the predictive ability of the models.

The findings from this case study fit inside the framework established by Falk et al. [16] and Som et al. [15], and broadly agree with their findings. However, for us, SSNdesign streamlined the process of discovering these results. The same code sufficed for both the CPD- and K-optimality experiments, with only a few lines' difference to account for the change in utility function (S2 Appendix). If required, we could easily have changed the covariance function underpinning the spatial relationships in the Lake Eacham network or the priors on the covariance parameters. SSNdesign will enable aquatic scientists and statisticians to construct designs for their own monitoring programs or make decisions about them with ease. Bespoke code will no longer be required, expanding access to the sophisticated methodologies of optimal and adaptive design.

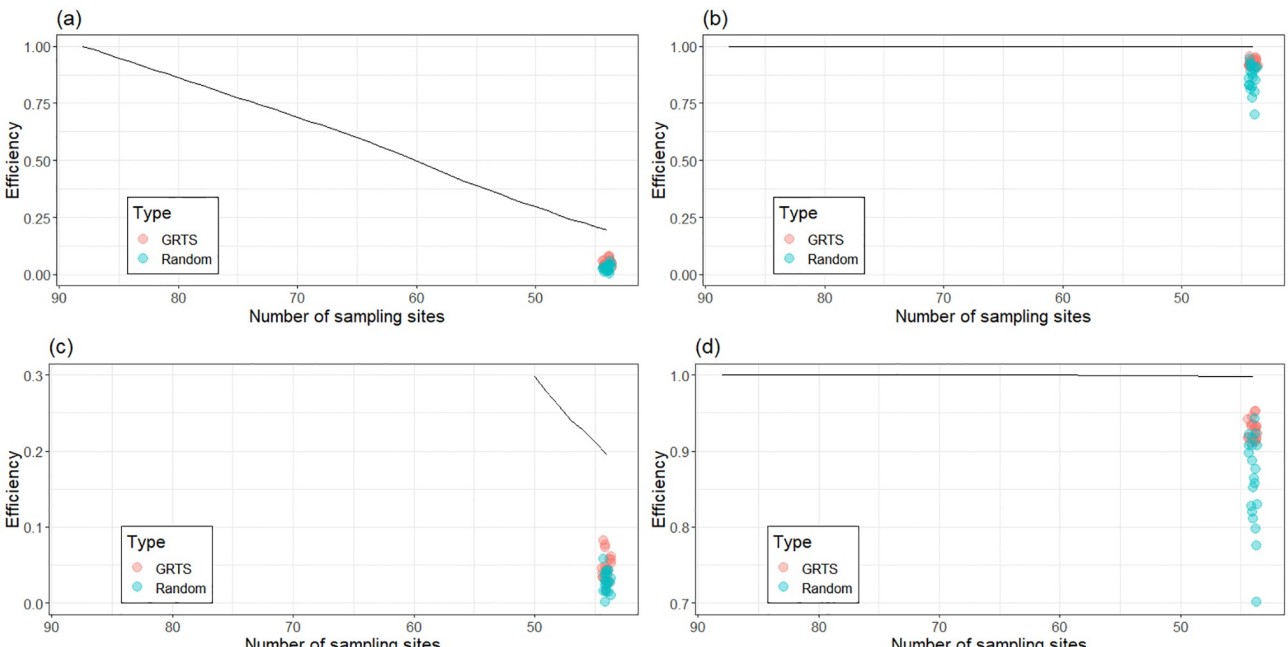

**Fig 4. Information loss in our optimal designs as we remove sites one-by-one using the (a, c) CPD-optimal and (b, d) K-optimal utility functions.**
The panels (a) and (b) have axes for efficiency fixed between 0 and 1. Panel (c) is zoomed on the y-axis range 0-0.3 and panel (d) on the y-axis range 0.7-1.0. Efficiency represents a ratio of the expected utilities of each design and the full 88 site design. The black line indicates the efficiency of the optimal design with a certain number of sampling sites. The 20 red dots and 20 blue dots in each panel represent the efficiencies of 44-site GRTS and random designs, respectively. These serve as a baseline measure of comparison for the optimal design, which should have higher relative efficiency. We only compare the efficiencies of the GRTS, random and optimal designs when there are 44 sampling sites in the design because the 44 site monitoring program is the final result.

## Case study 2: Pine River

In the second case study, we demonstrate how additional sites can be selected to complement the information provided by a set of legacy sites using simulated data. The objective of the adaptive design process is to generate a design that can be used to accurately predict dissolved oxygen over the stream network at unsampled locations.

We did not actually have data at legacy sites. Therefore, for this example, we started by simulating four years of maximum daily dissolved oxygen (DO, mg/L) data at 900 locations throughout the river network. We then selected 200 of these sites using a GRTS design, which were treated as legacy sites. The first two years of data from the legacy sites were treated as historical data and used to estimate the fixed effects and covariance parameters using a spatial statistical stream-network model (S2 Appendix). Five random starts were used to find an adaptive design which maximises the K-optimality utility function (Table 1). We estimated $U(d)$ using $M = 500$ Monte-Carlo draws from our independent log-normal priors on the covariance parameters. The results were used to select an additional 50 sites in year 3; after which the model was refit to the full dataset and an additional 50 sites were selected in year 4. Thus, the final dataset included 300 sites, giving rise to 950 observed DO measurements collected across four years (Fig 5). The result is shown in Fig 6.

We validated the adaptive design by computing its relative efficiency compared to 20 GRTS and 20 random designs of the same size. The GRTS designs were sequentially constructed using the 'master sample' approach [35]. We compared relative efficiency by computing the sum of the kriging variances for the same 900 prediction sites that were used when optimising

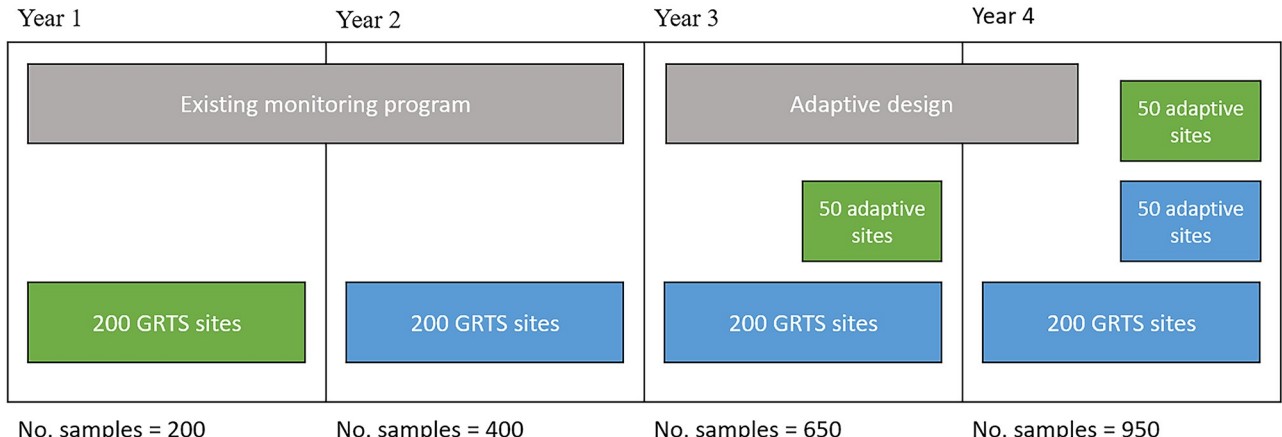

**Fig 5. A schematic diagram showing the adaptive design process over four years.** Green rectangles indicate new sites (i.e. sampling locations) that have been added to the monitoring program, and blue rectangles indicate that sites have been retained from previous years.

the design. Note that the sum of the kriging variances is simply the inverse of the expected utility. For the purpose of validation, this expected utility was computed using 1000 prior draws to ensure our approximations to the expected utility were accurate. We did not include the 200 fixed GRTS sites in the validation procedure because we wanted to assess whether there were

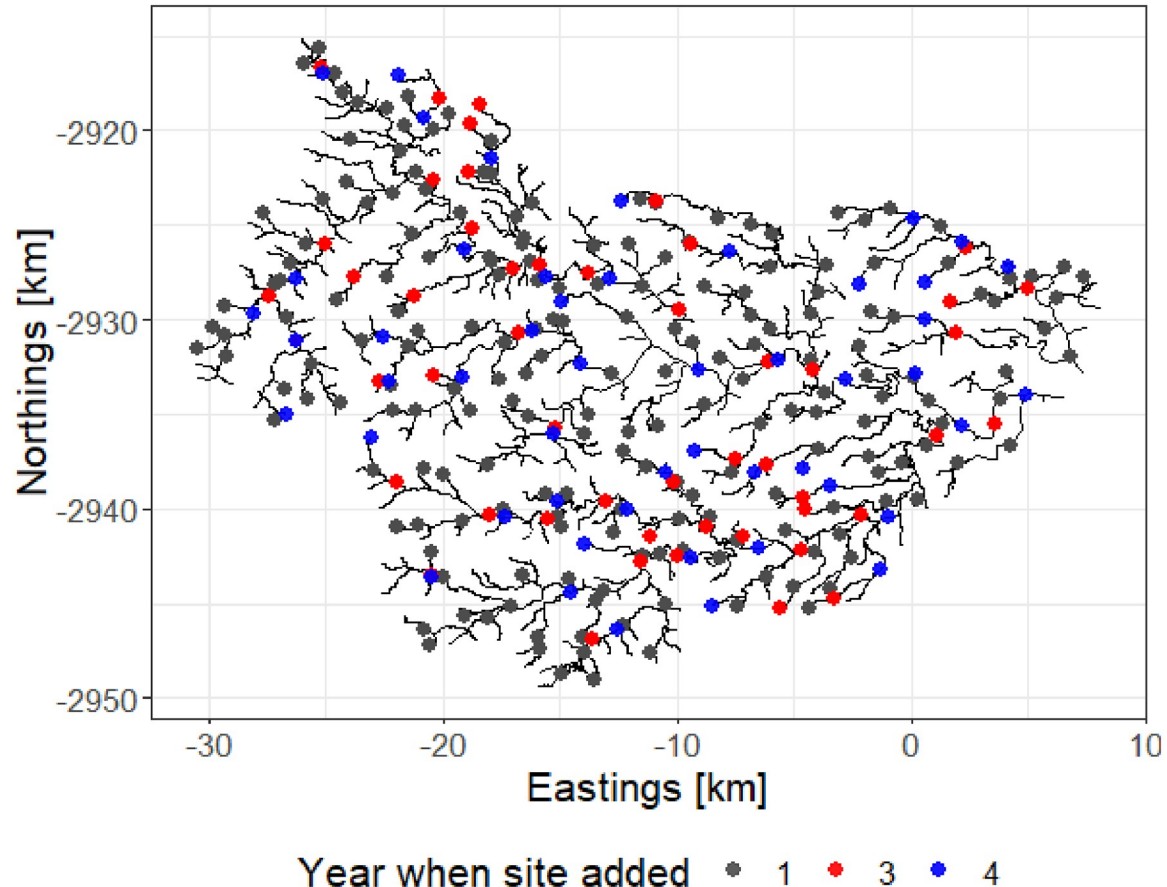

**Fig 6. The Pine River stream network.** The sites on the network represent the evolution of the adaptive design through time.

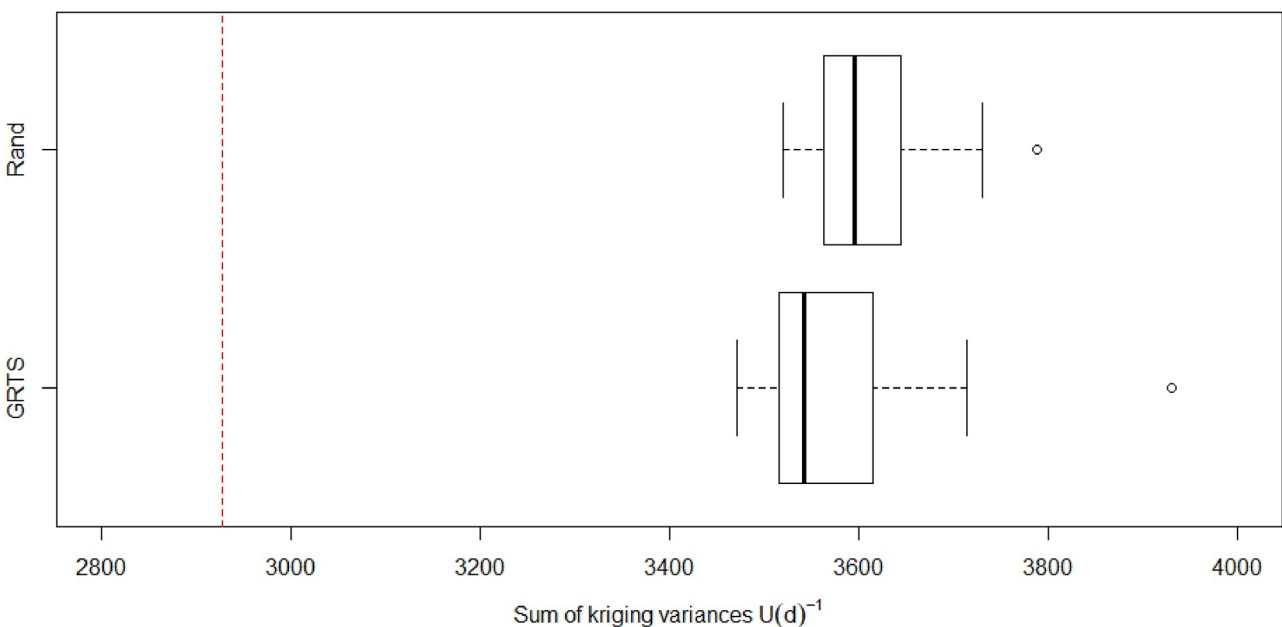

**Fig 7. The sum of kriging variances for both Generalised Random Tessellation Sampling (GRTS) and random designs computed using 1000 draws from the priors set on the covariance parameters.** The dashed red line represents the sum of the kriging variances for the adaptive design. The performance of the adaptive design is plotted this way as opposed to as a boxplot because there is only one adaptive design.

any additional benefits gleaned from the adaptive design. Thus, the efficiencies for each design (i.e. adaptive, GRTS, and random) were based on the 150 measurements collected at 100 locations in years 3 and 4.

The results showed that the adaptive design was more efficient than both the GRTS and random designs, which had 74.5-84% relative efficiency. In terms of variance units, using the adaptive design reduces the total variance across 900 prediction sites by approximately 545-1004 variance units compared to GRTS and random designs (Fig 7). This demonstrates that adaptive design represents a far better investment in terms of predictive ability than designs formed without optimisation.

## Conclusions

The SSNdesign package brings together a large body of work on optimal spatial sampling, pseudo-Bayesian experimental design, and the complex spatial data processing and spatial statistical modelling of stream network data. We demonstrate how the package can be used in two contexts which should prove useful for scientists and managers working in stream ecosystems; particularly where monitoring programs lack the resources to comprehensively sample the network, but must nevertheless estimate parameters for complicated spatial processes and accurately predict in-stream conditions across broad areas.

Compared with other packages for spatial sampling, such as geospt [25] and spsurvey [23], SSNdesign represents a significant advance in functionality for stream network data. The SSNdesign package integrates directly with the data structures, modelling and model diagnostic functions from a well-established R package for streams, SSN [20]. As a result, the hydrological distance measures and unique covariance functions for streams data can also be used in the design process. This cannot be accomplished using other packages for spatial design, which are restricted to using Euclidean distance in the conventional 2-dimensional domain of geostatistics. SSNdesign has been written specifically to deal with problems of

model-based design (i.e. obtaining the best information from a model given that a spatial statistical model for in-stream processes can be specified). This distinguishes SSNdesign from packages such as spsurvey [23], which selects designs based on factors such as the inclusion probability for a site given characteristics of the underlying stream segments. However, note that we also include tailored functions to simplify the process of generating designs for stream networks as per Som et al. [15], which include designs that can be formed using spsurvey.

In addition to the existing functionality in SSNdesign, there are opportunities for further development. In particular, the authors of the SSN package [20] are working on extending its functionality to include computationally efficient models for big data (Personal Comm., J. Ver Hoef) and we expect there will be major performance boosts for the empirical utility functions where spatial stream network models must be fit iteratively. Computationally efficient spatial stream network models also open up possibilities to include new functionality within the SSNdesign package. For example, users must currently specify a single spatial stream network model as the 'true' model underpinning the design problem. This raises several important questions about the possibility of true models, and the uncertainty about which of several plausible, competing models ought to be chosen as the true model. Increased computational efficiency would allow us to implement a model-averaging approach [36] so that users will be able to specify several plausible models as reasonable approximations for the true model. Expected utilities for designs will then be averages of the expected utilities of the design under each plausible model. This approach would make optimal and adaptive designs more robust because the averaging mitigates the possibility that designs are being chosen to obtain the best information about the wrong model. Our hope is that managers of freshwater monitoring programs can more efficiently allocate scarce resources using optimal and adaptive designs for stream networks, which we have made accessible through the SSNdesign package.

## Supporting information

**S1 Appendix. Background on optimal and adaptive pseudo-Bayesian design.**
(PDF)

**S2 Appendix. SSNdesign—An R package for pseudo-Bayesian optimal and adaptive sampling designs on stream networks.**
(PDF)

**S3 Appendix. Glossary of terms.**
(PDF)

## Acknowledgments

Computational resources and services used in this work were provided by the eResearch Office, Queensland University of Technology, Brisbane, Australia.

The findings and conclusions in this article are those of the authors and do not necessarily represent the views of the US Fish and Wildlife Service nor the National Marine Fisheries Service, NOAA. Any use of trade, product, or firm names is for descriptive purposes only and does not imply endorsement by the US Government.

## Author Contributions

**Conceptualization:** Alan R. Pearse, James M. McGree, Paul Maxwell, Erin E. Peterson.

**Data curation:** Paul Maxwell, Erin E. Peterson.

**Formal analysis:** Alan R. Pearse.

**Funding acquisition:** James M. McGree, Erin E. Peterson.

**Investigation:** Alan R. Pearse.

**Methodology:** Alan R. Pearse, James M. McGree, Nicholas A. Som, Catherine Leigh, Jay M. Ver Hoef, Erin E. Peterson.

**Software:** Alan R. Pearse, Nicholas A. Som, Jay M. Ver Hoef, Erin E. Peterson.

**Supervision:** James M. McGree, Erin E. Peterson.

**Validation:** Alan R. Pearse, James M. McGree, Catherine Leigh, Paul Maxwell, Jay M. Ver Hoef, Erin E. Peterson.

**Visualization:** Alan R. Pearse, Erin E. Peterson.

**Writing – original draft:** Alan R. Pearse.

**Writing – review & editing:** Alan R. Pearse, James M. McGree, Nicholas A. Som, Catherine Leigh, Paul Maxwell, Jay M. Ver Hoef, Erin E. Peterson.

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
