## [Decision Letter · Decision Letter 0]

9 Jan 2020

PONE-D-19-26876

SSNdesign – an R package for pseudo-Bayesian optimal and adaptive sampling designs on stream networks

PLOS ONE

Dear Dr Pearse,

Thank you for submitting your manuscript to PLOS ONE. After careful consideration, we feel that it has merit but does not fully meet PLOS ONE’s publication criteria as it currently stands. Therefore, we invite you to submit a revised version of the manuscript that addresses the points raised during the review process.

We would appreciate receiving your revised manuscript by February 20, 2020. To enhance the reproducibility of your results, we recommend that if applicable you deposit your laboratory protocols in protocols.io, where a protocol can be assigned its own identifier (DOI) such that it can be cited independently in the future. For instructions see: http://journals.plos.org/plosone/s/submission-guidelines#loc-laboratory-protocols

We look forward to receiving your revised manuscript.

Kind regards,

Andreas C. Bryhn

Academic Editor

PLOS ONE

Journal Requirements:

Additional Editor Comments (if provided):

Dear authors,

Thank you for a well-written manuscript, Both reviewers recommended a minor revision and I agree with them. Again, I would like to apologise for the long review process, which was due to the difficulty to find suitable reviewers who agreed to participate. Please make sure to address all of the reviewers' comments in your rebuttal letter.

Reviewers' comments:

Reviewer's Responses to Questions

**Comments to the Author**

1. Is the manuscript technically sound, and do the data support the conclusions?

Reviewer #1: Yes

Reviewer #2: Yes

2. Has the statistical analysis been performed appropriately and rigorously? 

Reviewer #1: Yes

Reviewer #2: Yes

3. Have the authors made all data underlying the findings in their manuscript fully available?

Reviewer #1: Yes

Reviewer #2: Yes

4. Is the manuscript presented in an intelligible fashion and written in standard English?

Reviewer #1: Yes

Reviewer #2: Yes

5. Review Comments to the Author

Reviewer #1: This paper presents a new R statistical computing package called "SSNdesign" for obtaining statistically optimal (according to a range of different criteria) sampling designs on stream networks. As such, it provides a useful tool for scientists who manage freshwater ecosystems. The package takes advantage of recently (within the last 10-15 years) developed methodology for statistically modeling spatially correlated data taken at points on a stream network.

The paper describes the software package at an appropriate level of detail, and the two case studies are nice and (in my opinion) need no improvement. I do not have expertise on the computational aspects (storage requirements, speed, reliability) of the package so I will limit my comments to those things on which I do have expertise, namely the relevant statistical models and utility functions, as presented in Supplement 1. First, however, here are a few minor comments referring to specific line numbers in the paper itself.

Line 80: Change "pseudo-Baeysian" to "pseudo-Bayesian"

Line 100: Change "Tesselation" to "Tessellation"

Line 200: Change "before" to "then" (so that the sentence reads more smoothly)

Line 279: Delete "based"

Line 298: Delete the period after "distribution"

Line 346: Change "dissovled" to "dissolved"

Now here are some comments on Supplement 1, with references to its line numbers.

Line 32: Remove the comma between "second-order" and "stationary"

Lines 32-33: It is a bit of an oversimplification to say that the moving average models produce covariance functions that depend on "the separation distance between two locations" because tail-down covariance functions generally are functions of two distances (distance of each location to their common confluence) rather than merely the separation distance between locations. So this sentence needs a minor correction.

Line 34: It is somewhat imprecise to refer to a point at which a function "starts." What you mean is that the moving average function is nonzero (positive, in fact) only at and upstream from a point. So you should change this sentence by changing "starts at some location and is nonzero only upstream from that location" to "is nonzero only upstream from a location." Make a similar edit in line 37 when describing the tail-down covariance function.

Line 44: Change "set" to "vector"

Line 67: When two sites are flow-unconnected, a and b, as defined, ALWAYS sum to h (the stream distance between the two sites). Thus, just change this sentence by deleting the phrase "when a+b=h"

Lines 71-78 and expression (5): For consistency with expression (4), use bold type for Y, X, etc.

Line 72: Here it should be noted that X is assumed to have full column rank; otherwise, the matrix in parentheses in line 158 is not invertible.

Line 94: Change "join" to "joint"

Line 106: Change "funciton" to "function"

Lines 120-133: As this is written it is unclear (1) what is the "first" site in d_o, and (2) whether the design with the largest utility (among the N-n designs) or an arbitrary design among the N-n designs is the design that replaces d^*. Please clarify.

Line 164: Because you gave an expression for the information matrix associated with the covariance parameters in expression (9), it would be good to give an expression for the information matrix associated with the fixed effect parameters here as well.

Reviewer #2: Comments for the author have been attached

6. PLOS authors have the option to publish the peer review history of their article (what does this mean?). If published, this will include your full peer review and any attached files.

Reviewer #1: No

Reviewer #2: No

---

## [Author Response · Author response to Decision Letter 0]

28 Jul 2020

Response to reviewers

Reviewer 1

The reviewer’s comments were that

This paper presents a new R statistical computing package called "SSNdesign" for obtaining statistically optimal (according to a range of different criteria) sampling designs on stream networks. As such, it provides a useful tool for scientists who manage freshwater ecosystems. The package takes advantage of recently (within the last 10-15 years) developed methodology for statistically modeling spatially correlated data taken at points on a stream network.

The paper describes the software package at an appropriate level of detail, and the two case studies are nice and (in my opinion) need no improvement. I do not have expertise on the computational aspects (storage requirements, speed, reliability) of the package so I will limit my comments to those things on which I do have expertise, namely the relevant statistical models and utility functions, as presented in Supplement 1. First, however, here are a few minor comments referring to specific line numbers in the paper itself.

The reviewer then provides a list of line-by-line corrections for small matters (spelling, grammar, etc.). 

Firstly, in response, we thank Reviewer 1 for their constructive and positive feedback on our manuscript. Secondly, we list our responses/changes below against the line-by-line commentary of the reviewer. The reviewer’s comments are shown in bold font. 

Line 80: Change "pseudo-Baeysian" to "pseudo-Bayesian"

Done.

Line 100: Change “Tesselation” to “Tessellation”

Done.

Line 200: Change “before” to “then” (so that the sentence reads more smoothly)

Done.

Line 279: Delete “based”

Done.

Line 298: Delete the period after “distribution”

Done.

Line 346: Change “ issolved” to “dissolved”

Done.

Now here are some comments on Supplement 1, with references to its line numbers.

Line 32: Remove the comma between "second-order" and "stationary"

Done. 

Lines 32-33: It is a bit of an oversimplification to say that the moving average models produce covariance functions that depend on "the separation distance between two locations" because tail-down covariance functions generally are functions of two distances (distance of each location to their common confluence) rather than merely the separation distance between locations. So this sentence needs a minor correction.

We agree with the reviewer. This now reads 

…they can be described by a mean function that depends on the location within the network, and a second-order stationary covariance function. Traditional covariance functions parameterise the dependence between observations in terms of the Euclidean distance separating two locations, but this is less straightforward in the context of stream networks. Stream network covariance functions and the distance metrics they use may depend on flow connectivity. Details on these covariance functions are provided below.

Line 34: It is somewhat imprecise to refer to a point at which a function “starts.” What you mean is that the moving average function is nonzero (positive, in fact) only at and upstream from a point. So you should change this sentence by changing “starts at some location and is nonzero only upstream from that location” to “is nonzero only upstream from a location.” Make a similar edit in line 37 when describing the tail-down covariance function.

Done.

Line 44: Change "set" to "vector"

Done.

Line 67: When two sites are flow-unconnected, a and b, as defined, ALWAYS sum to h (the stream distance between the two sites). Thus, just change this sentence by deleting the phrase "when a+b=h"

Done.

Lines 71-78 and expression (5): For consistency with expression (4), use bold type for Y, X, etc.

We have decided to drop the bold type for Y, X in Expression (4). This makes this expression consistent with the rest of the appendix and the main manuscript, where bold font is not used for Y, X.

Line 72: Here it should be noted that X is assumed to have full column rank; otherwise, the matrix in parentheses in line 158 is not invertible.

Done. The sentence now reads “… X is a design matrix with full column rank for the fixed effects”.

Line 94: Change "join" to "joint"

Done.

Line 106: Change "funciton" to "function"

Done. 

Lines 120-133: As this is written it is unclear (1) what is the "first" site in d_o, and (2) whether the design with the largest utility (among the N-n designs) or an arbitrary design among the N-n designs is the design that replaces d^*. Please clarify.

In response to (1), the initial randomly-selected design can be thought of as a set of sites d_0={s_1,s_2,s_3,…,s_n}. The “first site” in this design simply refers to s_1 in the set. We agree this could be unclear to a reader. The updated text is given beneath the next paragraph.

As for (2), the design with the largest utility among the N-n designs replaces d^* if and only if its utility is greater than the utility of d^*. That is, if d^b is the “best” design among the N-n designs evaluated at that iteration of the coordinate exchange algorithm, d^* is replaced by d^b if and only if U(d^b )>U(d^* ). Sometimes U(d^b) does not satisfy this condition because there is no design among the N-n recently evaluated designs that improves on the current best design. We agree this could be unclear to readers so we have updated the text to be:

In this work, we use a greedy exchange algorithm (Algorithm 2) to locate optimal designs (9,13). The greedy exchange algorithm works by optimising the choice of each of n sites one-by-one. Initially, a random design with n sites is proposed and becomes d_0 = {s_1 ,s_2 ,...,s_n } (the initial design) and d^* (the design which currently has the highest value of U(d)). From this point, we begin the coordinate exchange. Note that there are N - n candidate points not currently in d^*. The first site in d_0 (s_1) is then swapped out for each of the N-n candidate sites. The expected utilities of the resulting designs are recorded. If any designs have an expected utility larger than U(d^* ), the design with the highest expected utility replaces d^*. Then we update our pool of candidate sites, and we begin to exchange the next site. Otherwise, the design reverts to d^* and the next site in the design is exchanged for each candidate site. 

Line 164: Because you gave an expression for the information matrix associated with the covariance parameters in expression (9), it would be good to give an expression for the information matrix associated with the fixed effect parameters here as well.

We agree with the reviewer here. Thank you for catching this oversight. The expression for I(d,β_gls) is now given in the expanded inline at Line 167.

Reviewer 2

 Case study 1 

 In this paper, case study one (Lake Eacham) is based on an existing dataset (i.e. Falk 2014 uses this dataset as well) and the underlying methodology used for the study is not new (optimal experimental design was used in Falk 2014 and Som 2014). I think it's important to state this clearly and cite the above-mentioned prior work. 

We agree that we should have more prominently cited and acknowledged these previous works. However, we note the Lake Eacham dataset was originally published on the U.S. Forest Service website ( https://www.fs.fed.us/rm/boise/AWAE/projects/SSN_STARS/software_data.html) as an example data set for the software package STARS (Peterson, 2011):

Peterson EE (2011). STARS: Spatial Tools for the Analysis of River Systems – A Tutorial. Technical Report EP111313, Commonwealth Scientific Industrial Research Organisation (CSIRO). URL: http://www.fs.fed.us/rm/boise/AWAE/projects/SSN_STARS/software_data.html#doc.

We have adjusted the text of the first two paragraphs of Case Study 1 to reflect the origins of the data set and to emphasise the previous work done by Som et al. (2014) and Falk et al. (2014). 

 Furthermore, the case study is currently presented like a vignette (an application of the SSNdesign package to real data), but doesn't make clear how it is relevant in demonstrating the contribution of the SSNdesign package. Given that both the dataset and the methodology in this case study were already presented in prior work, the majority of content of the case study is better suited for an R package vignette or as supplementary material. Instead, I think the paper would be better served by giving (i) a more concise overview of the case study using SSNdesign, with emphasis on how the software improves the ability to do such analysis, and (ii) a discussion of how the SSNdesign package better addresses the experimental design problem compared with alternatively available software like spsurvey or geospt. 

Though the reviewer is correct that neither the Lake Eacham dataset nor optimal design problems on stream networks are novel, we must point out that the following elements of the case study differentiate it from previously published examples with these data:

 The CPD-optimality utility function was not used in either Falk et al. (2014) or Som et al. (2014). At best those two papers used the C-optimality and D-optimality utility functions separately. 

 Our method of dropping sites one-by-one from the design was not used in Falk et al. (2014). Consequently, the insights we gain from tracking information loss over the successive designs are novel. 

To address the point that the case study reads like a vignette, we would like to first point out that, in some ways, it is meant to do so. This paper documents our R package SSNdesign and showcases its capabilities. We chose the case studies to be of interest to a wide audience of readers and both address common (applied) design challenges that aquatic managers face; noting that novel methods are also presented in Case Study 2. The code used in Som et al. (2014) and Falk et al. (2014) were not made publicly available and, as a result, optimal and adaptive design methods on stream networks have remained inaccessible to the vast majority of potential users over the last six years. Thus, our main purpose is simply to demonstrate how the SSNdesign package simplifies previously complex tasks that required bespoke code to be developed. 

At the same time, we agree with the reviewer that we would benefit shifting the focus of our discussion of Case Study 1. We have edited the discussion of the first case study to highlight the utility of SSNdesign in producing these results, and we have also highlighted the connections between this case study and Falk et al. (2014). 

At Lines 289-290, we have entered:

In a similar experiment to that of Falk et al. (21), we used optimal experimental design to reduce the number of survey sites by half, …

At Lines 344-352, we have added the following paragraph of discussion:

The findings from this case study fit inside the framework established by Falk et al. (21) and Som et al. (20), and broadly agree with their findings. However, for us, SSNdesign streamlined the process of discovering these results. The same code sufficed for both the CPD- and K-optimality experiments, with only a few lines’ difference to account for the change in utility function (S2 Appendix). If required, we could easily have changed the covariance function underpinning the spatial relationships in the Lake Eacham network or the priors on the covariance parameters. SSNdesign will enable aquatic scientists and statisticians to construct designs for their own monitoring programs or make decisions about them with ease. Bespoke code will no longer be required, expanding access to the sophisticated methodologies of optimal and adaptive design.

Case study 2 

 Case study two (Pine River) is a new dataset (synthetically simulated) and the underlying methodology used for the study is new (the authors state that adaptive experimental design problems have not been used under the SSNM framework). This is a case study that supports the motivations of this paper. 

Yes, thank you.

In my experience, this R package will be an invaluable tool to researchers and managers in the freshwater monitoring field. I recommend this manuscript be published pending minor revisions. I look forward to reading a revised version of the manuscript.

Thank you for your insightful and positive feedback! 

Abstract Line 17 “and so effective and efficient survey designs…”: delete “and” (double conjunction).

Done.

Abstract Lines 21 - 23 “Thus, unique challenges of geostatistics and…”: This sentence doesn’t feel supported by the sentences which precede it. This sentence suggests that the unique challenges of geostatistics on stream networks motivated the development of SSNdesign. But, it would be more accurate to say that these challenges motivate the development of the methodology implemented in SSNdesign. A better motivation for an R package like SSNdesign would be the lack of available software that provides access to, or application of, these methods. For example, a sentence summarizing the lack experimental design R packages which account for the unique SSN structure would be stronger motivation (i.e. see 5th paragraph, lines 94 - 109 of the introduction).

We agree with the reviewer’s assessment. We have therefore modified this part of the abstract to read:

Geostatistical models for stream network data and their unique features already exist. Some basic theory for experimental design in stream environments has also previously been described. However, open source software that makes these design methods available for aquatic scientists does not yet exist. To address this need, we present SSNdesign, ...

Lines 58 - 61: Here the authors have demonstrated the increased use of SSNMs in the literature. The end of this paragraph would be a good place to cite other papers which empirically looked at experimental design utilizing SSNMs. I think it would strengthen the paper’s motivation and further highlight the research needs of an R package which tackles these questions. For example Marsha et al, 2018, Monitoring riverine thermal regimes on stream networks: Insights into spatial sampling design from the Snoqualmie River, WA, Ecological Indicators 84: 11-26. Some of the takeaways from these papers align with those learned from the case studies you have included.

We agree with the reviewer on this point. Being able to point to specific studies focussing on experimental design on stream networks would indeed strengthen our case studies. Thank you for pointing us to this useful reference. We have incorporated and cited this paper on Lines 61 - 62:

see, for example, Isaak et al. (11) and Marsha et al. (12), both of whom model thermal regimes in streams. Marsha et al. (12) further consider questions of site placement and sample size based on their data.

Line 67 “Utility functions are mathematical…”: This sentence feels out of place here. Suggest moving it to line 70 before “A variety of utility functions are available…”

We agree that this sentence reads a bit awkwardly, but we also thought carefully about where to introduce this concept. We believe it is important to keep it in place because it defines what utility functions are in general before the following sentence, which is about differing design objectives affecting the spatial distribution of samples. 

Line 81 “which measure the suitability of an experimental design for some purpose”: The authors already defined a utility function in the previous paragraph. I suggest deleting this and continue with “often depend on…”

Done.

Line 87 “In this paper and in SSNdesign…”: Upon first reading I don’t understand the separation between “this paper” and “SSNdesign.” By this paper do you mean application to the case studies? Perhaps make this clearer.

On further reflection, we agree with the reviewer that this is confusing. In fact, separating the two is unnecessary. Therefore, we have cut out the reference to the paper and now the sentence reads “In SSNdesign, we use the pseudo-Bayesian approach.”

Figure 1: 

 It is not clear where the reader should be starting with this flowchart. My experience with using SSNMs in R indicate to me how to interpret this flowchart, but I can see how a reader who is new to working with SSNMs in R could be confused. A flow chart which is left justified (or column aligned) and which then flows in parallel would be clearer (like Figure 2). 

Thank you for this feedback. We agree the intended reading of the flowchart is not clear. We also agree with your suggestions on how to improve it. Please see the response to your comment below on specifically how we went about this.

 When interpreting this flowchart it is not clear if the two “.ssn folder” objects are the same object fundamentally. If they are, then the reader wonders why one is required to be run through the importSSN function before SimulateOnSSN, while the other can go straight to SimulateOnSSN. My interpretation is that the bottom path “createSSN → .ssn folder → SimulateOnSSN” represents a continued R session (the scenario where the user starts with nothing); while the other path represents a new R session where the user starts with a “.ssn folder” object. If this interpretation is correct, I would suggest splitting this figure into two sections (one right and one left; or one top and one bottom) illustrating the two scenarios. 

Again, thank you for this feedback. To clarify, 

 The 2x “.ssn folder” do not refer to the same object, just to a data structure that the two workflows have in common. (i.e. both use .ssn folders to store the stream network data.) 

 The “.ssn folder” appearing in the top workflow refers to a pre-existing dataset where the user has, at a minimum, a set of stream edges in a shapefile. From here, if the user has no observed data in a sites shapefile, then they can simulate observed and prediction locations using generateSites. Once the observed and prediction locations have been generated, the user may then simulate observed response values on those sites using SimulateOnSSN. Alternatively, if the user has a .ssn folder that already contains an observed sites and prediction locations shapefile(s), then they can go straight to simulating response values if necessary.

 The “createSSN -> “.ssn folder” -> SimulateOnSSN” refers to a situation where the user has no data for a stream network. That is, they must artificially generate all parts of the stream dataset including the stream edges and sites. The createSSN function is responsible for creating a new .ssn folder that contains the completely simulated stream network. These data are then passed on to the SimulateOnSSN function to provide simulated response values for the simulated survey sites.

We have updated Figure 1, incorporating your suggestions for improving readability. Specifically, we have disentangled each of the three cases from one another. There are now three rectangles stacked vertically, each one representing a different scenario (i.e. whether you have stream edges and survey data, just stream edges, or no GIS or survey data at all). We believe the new Figure 1 is a vast improvement on the original. 

The Expected utility estimation and maximisation section: In general these two paragraphs are concise and very well explained. 

Thank you!

The Utility functions for optimal and adaptive experimental designs section: 

 Reading about the adaptive design technique in SSNdesign, I see that maximizing U(d | d0:t-1, y0:t-1) depends upon d0 (an initial model design). This naturally makes the reader wonder the sensitivity of the initial model chosen. Did the authors look at how the choice of d0 affects future iterations, or even the final outcome, in their adaptive design? Or are there any other studies that looked at this that can be cited? 

This is an insightful question. The choice of d0 will affect design choices in future iterations. This is because initially, prior information about the model parameters is formed based on d0. If this is vague, then it is likely that the iteratively chosen designs will become more informative as we become more certain about the parameter values. In contrast, if the prior information is initially quite informative, then the iteratively chosen designs should start out being relatively informative. The effect that the choice of d0 has on subsequent designs was not formally investigated via simulation as typically (at least in the stream modelling scenarios we have encountered) d0 will be quite uninformative, particularly for covariance parameters. Indeed, this really motivates the need to adopt adaptive design methods as, if the parameters are well estimated based on d0, then static (non-adaptive) design approaches would probably be suitable.

 I see where the authors explicitly discuss solving adaptive design problems (paragraph 2), but where are optimal design problems discussed in this section? 

The main purpose of this section is to provide concise definitions of all the utility functions that we have included in SSNdesign. The main focus of this section is Table 1 where we write out the equations for the utility functions; describe their arguments; and their intended use-cases (e.g. for selecting covariance/fixed effects parameters, or for minimising the average kriging standard errors across prediction sites). However, detailed discussions of both the optimal and adaptive design procedures are actually provided elsewhere (see the section entitled “Expected utility estimation and maximisation”, as well as the S1 Appendix). We explicitly discuss the myopic design approach in paragraph 2 of this section (while also pointing readers to the relevant sections and algorithms in the S1 Appendix for more detailed information) because it is critical information for understanding the utility functions we have included in the package for adaptive design. Utility functions for myopic designs look different to those used in backward induction. Therefore, we believe this brief discussion of the myopic design approach is necessary here. 

Line 195 “Space-filling designs are designs which…”: delete “are designs which, ideally” so it reads “Space-filling designs contain roughly…” 

Done.

Line 201 “Here we used an algorithm…”: change “used” to “use” 

Done.

Line 219 “Users may also define…”: change to “Users may also define their own utility functions since the optimiseSSNDesign function has the flexibility to…” It doesn’t make sense to use a function to define another function.

Done.

Line 258 “Field data collection can be expensive and so existing…”: delete “and” (double conjunction). 

Done.

Lines 271 - 275: Just a formatting note to capitalize “Identify” and “Set” as the beginning of each list point. 

Done.

Line 274 “Set priors that specify what the…”: I suggest changing this sentence as follows, “Set priors that specify the likely relationship between covariates and the response based on expert opinion and/or a literature review.” This feels more concise and clear. 

Done, but we have added “previously collected data” so that the sentence reads:

Set priors that specify the likely relationship between covariates and the response based on previously collected data, expert opinion and/or a literature review.

Line 281: This is a previously published dataset that should be cited. 

Yes, we have now cited it correctly as

Peterson EE (2011). STARS: Spatial Tools for the Analysis of River Systems – A Tutorial. Technical Report EP111313, Commonwealth Scientific Industrial Research Organisation (CSIRO). URL: http://www.fs.fed.us/rm/boise/AWAE/projects/SSN_STARS/software_data.html#doc.

Figure 4: This is a good figure which effectively illustrates the author’s conclusions from case study 1. 

Thank you! 

 I suggest the two plots use the same y-axis scale (0.0 – 1.0). It could be misleading to the reader otherwise. 

Yes, we have done this for the updated version of Figure 4. However, we have expanded the number of panels to four. The reasons for this are partially explored below and further explained in our response to the reviewer’s next point. 

The updated plot now looks like this:

The following outlines how this figure works:

 Panels (a) and (d) are the panels (a) and (b) respectively from the original Figure 4. 

 Panels (a) and (b) in this version have their y-axes scaled to 0.0-1.0. The fact that both plots have the same axis scales means that it is possible to make direct comparisons of the two plots side-by-side. 

 Panels (c) and (d) are zoomed into the regions of the y-axes that allow the variability in the GRTS and random designs to be more clearly displayed. The total range of the y-axis is the same for both panels (0.3) but panel (c) shows the region of the plot with the y-axis fixed between 0.0-0.3 whereas panel (d) has its y-axis fixed between 0.7-1.0. 

 The feature in this figure that stands out to me the most is the lack of variability in efficiency under the GRTS designs using the CPD-optimal utility function (a) and the larger variability under the GRTS designs using the K-optimal utility function (b). Do the authors have any ideas about why that would be?

The problem that the reviewer has described here is an artifact of the inconsistent y-axis scales in the original Figure 4. The efficiencies of GRTS designs exhibit similar levels of variability under the CPD- and K-optimality utility functions. As it is now clear from panel (c) in the updated Figure 4, the GRTS designs under CPD-optimality exhibit efficiencies approximately in the range of 0.03-0.09 (total range = 0.06). Similarly, under K-optimality, the GRTS designs have efficiencies between 0.90-0.96 (total range = 0.06). We believe our updated Figure 4 no longer risks misleading readers in this way. 

Line 382 “Compared to other packages…”: change to “Compared with other packages…” Typical convention is that “compared to” is used for discussing similarities between fundamentally different objects while “compared with” is for discussing differences between two fundamentally similar objects. 

Done.

References

Falk MG, McGree JM, Pettitt AN. Sampling designs on stream networks using the pseudo-bayesian approach. Environmental and Ecological Statistics. 2014;21:751–73.

Peterson EE (2011). STARS: Spatial Tools for the Analysis of River Systems – A Tutorial. Technical Report EP111313, Commonwealth Scientific Industrial Research Organisation (CSIRO). URL: http://www.fs.fed.us/rm/boise/AWAE/projects/SSN_STARS/software_data.html#doc.

Som NA, Monestiez P, Ver Hoef JM, Zimmerman DL, Peterson EE. Spatial sampling on streams: Principles for inference on aquatic networks. Environmetrics. 2014;25(5):306–23.

---

## [Editor Report · Decision Letter 1]

18 Aug 2020

SSNdesign – an R package for pseudo-Bayesian optimal and adaptive sampling designs on stream networks

PONE-D-19-26876R1

Dear Dr. Pearse,

We’re pleased to inform you that your manuscript has been judged scientifically suitable for publication and will be formally accepted for publication once it meets all outstanding technical requirements.

Kind regards,

Andreas C. Bryhn

Academic Editor

PLOS ONE
---

## [Editor Report · Acceptance letter]

24 Aug 2020

PONE-D-19-26876R1 

SSNdesign – an R package for pseudo-Bayesian optimal and adaptive sampling designs on stream networks 

Dear Dr. Pearse:

I'm pleased to inform you that your manuscript has been deemed suitable for publication in PLOS ONE. Congratulations! Your manuscript is now with our production department. 

Kind regards, 

on behalf of

Dr. Andreas C. Bryhn 

Academic Editor

PLOS ONE